# Intimate physical contact between people from different households during the COVID-19 pandemic: a mixed-methods study from a large, quasi-representative survey (Natsal-COVID)

Pam Sonnenberg ,[1] Dee Menezes ,[2] Lily Freeman,[1] Karen J Maxwell ,[3] David Reid ,[1] Soazig Clifton ,[1,4] Clare Tanton ,[5] Andrew Copas ,[1] Julie Riddell ,[3] Emily Dema ,[1] Raquel Bosó Pérez ,[3] Jo Gibbs ,[1] Mary-Clare Ridge ,[1] Wendy Macdowall ,[6] Magnus Unemo ,[1,7] Chris Bonell ,[5] Anne M Johnson ,[1] Catherine H Mercer ,[1] Kirstin Mitchell ,[3] Nigel Field [1]

KM and NF are joint senior authors.

For numbered affiliations see end of article.

**Correspondence to**
Professor Pam Sonnenberg; p.sonnenberg@ucl.ac.uk

## ABSTRACT

**Objectives** Physical distancing as a non-pharmaceutical intervention aims to reduce interactions between people to prevent SARS-CoV-2 transmission. Intimate physical contact outside the household (IPCOH) may expand transmission networks by connecting households. We aimed to explore whether intimacy needs impacted adherence to physical distancing following lockdown in Britain in March 2020.

**Methods** The Natsal-COVID web-panel survey (July–August 2020) used quota-sampling and weighting to achieve a quasi-representative population sample. We estimate reporting of IPCOH with a romantic/sexual partner in the 4 weeks prior to interview, describe the type of contact, identify demographic and behavioural factors associated with IPCOH and present age-adjusted ORs (aORs). Qualitative interviews (n=18) were conducted to understand the context, reasons and decision making around IPCOH.

**Results** Of 6654 participants aged 18–59 years, 9.9% (95% CI 9.1% to 10.6%) reported IPCOH. IPCOH was highest in those aged 18–24 (17.7%), identifying as gay or lesbian (19.5%), and in steady non-cohabiting relationships (56.3%). IPCOH was associated with reporting risk behaviours (eg, condomless sex, higher alcohol consumption). IPCOH was less likely among those reporting bad/very bad health (aOR 0.54; 95% CI 0.32 to 0.93) but more likely among those with COVID-19 symptoms and/or diagnosis (aOR 1.34; 95% CI 1.10 to 1.65). Two-thirds (64.4%) of IPCOH was reported as being within a support bubble. Qualitative interviews found that people reporting IPCOH deliberated over, and made efforts to mitigate, the risks.

**Conclusions** Given 90% of people did not report IPCOH, this contact may not be a large additional contributor to SARS-CoV-2 transmission, although heterogeneity exists within the population. Public health messages need to recognise how single people and partners living apart balance sexual intimacy and relationship needs with adherence to control measures.

## Strengths and limitations of this study

► Natsal-COVID included a large, national sample and used quota-based sampling and weighting to achieve a quasi-representative population sample.

► The Natsal-COVID study was undertaken rapidly in response to the pandemic and benefited from a questionnaire design and approach developed by the team responsible for the decennial Natsal survey.

► Social desirability bias, especially in the context of the pandemic, may result in participants unwilling to report sensitive and prohibited behaviours, resulting in the underestimation of intimate physical contact outside the household (IPCOH).

► Although Natsal-COVID is a large and national sample, it is not a probability sample and web panels are likely to be less representative.

► Findings are likely to be broadly generalisable, but prevalence estimates should be interpreted with caution.

## INTRODUCTION

In the UK, as elsewhere, non-pharmaceutical interventions (NPIs) have been instituted to reduce the spread of SARS-CoV-2. These have commonly included physical distancing, particularly between people from different households. Mathematical models[1–3] to determine the effectiveness of NPIs are based on underlying parameters of population mixing and assumptions of adherence to NPIs. Internationally, there are a number of ongoing observational studies to estimate the frequency, type and duration of physical and social contact under different NPI scenarios to inform these models, as summarised in a

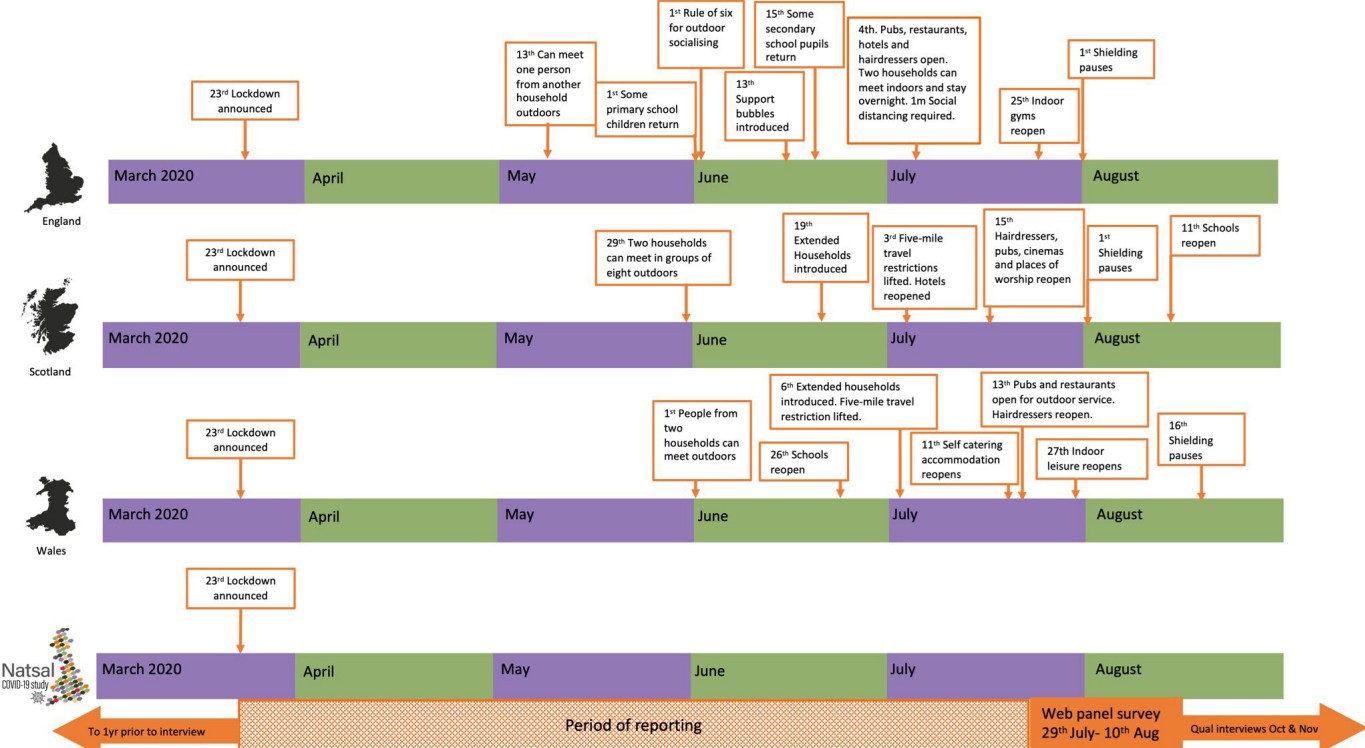

**Figure 1** Timeline of the COVID-19 pandemic in the UK, showing non-pharmaceutical interventions and the Natsal-COVID study.

recent review.[4] In the UK, these include the ONS Coronavirus and social impacts survey,[5] the Co-Mix study[6] and the COVID-19 social study.[7]

The COVID-19 pandemic has impacted all aspects of people's lives, including sexual behaviour and relationships.[8–14] However, no study to date has investigated in-depth the associations between sexual or romantic relationships or encounters and adherence to physical distancing measures,[4] and in particular, the occurrence of intimate physical contact outside the household (IPCOH) and associated factors. One web-based survey in Italy found that a quarter of participants who reported being sexually active during lockdown did not spend lockdown with their partner—suggesting that these individuals may have broken lockdown rules at the time.[14] The analysis did not look in-depth at characteristics or motivations of those participants. An understanding of the scale and circumstances of IPCOH is important because the nature of the contact is likely to increase risk of SARS-CoV-2 transmission and people meeting for sexual or romantic reasons may expand transmission networks by connecting households.

The first documented cases of COVID-19 in the UK were in January 2020, with a range of NPIs introduced over the following months (figure 1). Key dates include the announcement of the first national lockdown on 23 March 2020 when stringent measures were imposed and the public were instructed not to leave the house, except for buying food, emergency healthcare and travel between certain workplaces that were exempt (such as healthcare).

Subsequently, lockdown was eased, for example, in June 2020 measures to reduce social isolation among individuals living alone were introduced (referred to as 'support bubbles' in England, 'extended households' in Scotland and both terms were used in Wales).[15 16] The guidance in England stated that where there was only one adult in a household, this person could expand their support network to include one other household of any size. Those in a support bubble were able to meet indoors, be less than two metres apart and stay overnight as if they were members of the same household. Similar guidance applied in Scotland and Wales, although devolved nations had slightly different timescales. Throughout the UK, at no point between initial lockdown and the timing of the survey was IPCOH permitted.

Using data from the Natsal-COVID study, we estimated the proportion of adults living in Britain who reported IPCOH in the 4 weeks prior to the survey which took place during July–August 2020 and investigated associated factors. Through follow-up interviews with a sample of participants, we also aimed to understand the context, reasons and decision making around IPCOH.

## METHODS
### Study design
We carried out a web-panel survey (the Natsal-COVID study) of people aged 18–59 years living in Britain run by a survey research company (Ipsos MORI). Wave 1 data collection took place between 29 July 2020 and 10 August

2020 by Ipsos MORI. Broadly, the Natsal-COVID study aimed to understand sexual behaviour, sexual and reproductive health outcomes, and service use in the initial phase of COVID-19 and government responses, including NPIs.

## Study context

At the time of the survey, restrictions had started to relax in most parts of the UK, with ~10% of the population in local lockdown.[17] Data from the ONS household survey, used to measure trends in SARS-CoV-2 infections in the English population, showed that the percentage of people testing positive for SARS-CoV-2 was 0.40% in April 2020 and had decreased considerably to 0.06% by the end of June 2020.[18] Non-essential shops and hospitality were open, and communal worship had resumed. From early July, indoor mixing between households was allowed across the UK, although alongside physical distancing requirements, however, IPCOH was still not permitted.

## Participants and procedures

The target sample size was 6500 people comprising a core of 6000 aged 18–59, with a boost of 500 aged 18–29. Quotas were used with the aim of achieving a sample representative of the British general population by age, sex, region and social grade. Participants were recruited from established web panels which are run with stringent controls over recruitment and quality to ensure individuals can only join once. A total of 164074 panellists were sent an email invitation with recruitment continuing until quotas were reached. In addition, the dataset was weighted to the general population (by age, sex, ethnicity, social grade and sexual identity) to achieve a quasi-representative population sample. Further details of the survey design and methods, including sampling, recruitment, weighting, quality control and participant characteristics have been previously reported.[19] The full questionnaire is available at https://www.natsal.ac.uk/natsal-covid-study.[20] This included a set of questions about the type and circumstance of any romantic or sexual experiences in the 4 weeks prior to interview, and whether this was: with a person living in the same household; not living in the same household, but in the same bubble; or not living in the same household and not in the same bubble.

Qualitative interviews were conducted by telephone/video in October–November 2020 with a subsample of survey participants who reported IPCOH (n=741) of whom 63% (n=468) agreed to a follow-up interview. Eighteen respondents were included to fill age, gender, ethnicity and regional quotas, with the final sample characteristics as follows: male (n=9), female (n=9); 18–29 (n=5), 30–39 (n=5), 40–49 (n=4), 50–59 (n=4); white (n=14), from an ethnic minority group (n=4). Respondents came from across Britain and regions of the UK under different levels of lockdown represented (see figure 1).

## Statistical analysis

For the quantitative analysis, we used Stata's (V.16.1) complex survey analysis functions to incorporate weighting and stratification of the data. The denominator was the total sample (rather than only those sexually-experienced or sexually active) as this is key for modelling population impacts. Three categories of contact were defined: (1) holding hands/hugging/cuddling; (2) kissing and (3) oral/anal/vaginal sex or other genital contact, with the assumption that those in the latter categories also had the contact types in the former categories.

The primary outcome was any IPCOH with someone involving romance or sexual activity (hereafter referred to as a 'romantic/sexual partner'), irrespective of whether the person considered this to be within a bubble or not. Initial analyses indicated that splitting the outcome into three categories (no IPCOH, IPCOH inside a bubble, IPCOH outside a bubble) showed similar patterns to two categories (no IPCOH, IPCOH). This decision was also based on the small proportion of the population aged 16–59 living in single households who would be eligible to be in bubbles according to the definition. However, we included all three categories in our descriptive analysis of age groups and relationship status.

The explanatory variables included a range of demographic, behavioural and health-related factors hypothesised to impact IPCOH. Current relationship status was derived from questions on partnership type (single; steady/married/civil partnership; casual/new) and whether living together or not at the time of interview. To explore associations between IPCOH and risk behaviours, we include numbers of sexual partners in the past year, condomless sex, and alcohol consumption. Physical and mental health have been of concern during this pandemic, so, in addition to questions on general health and COVID-19 symptoms/diagnosis, depression and anxiety were analysed as factors associated with IPCOH. Participants were classified as having symptoms of depression or anxiety if they scored 3 or more on the Patient Health Questionnaire two item (PHQ-2) or Generalised Anxiety Disorder two item (GAD-2) scales.[21 22]

Natsal-COVID was inclusive in its approach to gender[19]; data are presented for all participants, and for men (including trans men) and women (including trans women). The 24 participants who identified 'in another way' are included when estimates are presented for 'all'. We present weighted proportions and 95% CIs. Using logistic regression, we estimated age-adjusted ORs (aORs). As age and relationship status are key factors in household structure, we conducted a secondary multivariable analysis (adjusting for age, gender and relationship status) and present aORs.

Qualitative data were analysed thematically, using framework analysis.[23] A coding framework was designed to explore respondents' motivations and decision making concerning IPCOH, and key themes were generated from the transcripts.

## RESULTS

Of the 6654 participants aged 18–59 years, 9.9% (95% CI 9.1% to 10.6%) reported IPCOH with a romantic/sexual partner in the previous 4 weeks. Of these, 86.2% reported oral/anal/vaginal sex or other genital contact, while the remainder reported only kissing (with or without holding hands/hugging/cuddling) (10.3%) or only holding hands/hugging/cuddling (3.5%).

Table 1 shows demographic, behavioural and health-related factors associated with IPCOH and figure 2 presents aORs for significant factors. For all factors examined, the AORs were broadly similar to the aORs and are shown in online supplemental table 1.

IPCOH varied by age, gender and ethnicity. The proportion of participants reporting IPCOH decreased with increasing age in men and women and was highest among those aged 18–24 (17.7%) (figure 3). Overall, women were less likely to report IPCOH than men (aOR 0.80, 95% CI 0.67 to 0.94). IPCOH was less likely to be reported among those of Asian/Asian British ethnicity (aOR 0.63; 95% CI 0.44 to 0.91) than those of white ethnicity. IPCOH was more likely among participants identifying as gay or lesbian (aOR 2.50; 95% CI 1.82 to 3.45) or bisexual (aOR 1.52; 95% CI 1.12 to 2.05) and these associations remained after adjusting for age, gender and relationship status (AOR 1.94; 95% CI 1.31 to 2.87 and AOR 1.50; 95% CI 1.00 to 2.25, respectively).

There were differences in reporting IPCOH according to current relationship status. The majority of participants (58.6%) were in steady co-habiting relationships and 2.5% of these people reported IPCOH. In contrast, a much smaller proportion of participants (7.2%) were in steady relationships but not living together, with over half (56.3%) of these people reporting IPCOH (aOR 49.4; 95% CI 37.2 to 65.5 compared with those in steady cohabiting relationships). A third of participants (29.4%) reported that they were single, but 8.9% of these people still reported IPCOH in the previous 4 weeks. IPCOH was higher (36.5%) in those who reported themselves to be in a casual/new relationship.

IPCOH was associated with reporting other risk behaviours, such as higher numbers of sexual partners in the past year (reaching 43.2% among those reporting three or more partners; aOR 5.60; 95% CI 4.19 to 7.46 compared with one partner) and condomless sex with one or more new partners in the past year (aOR 5.03; 95% CI 1.07 to 6.21 compared with not). IPCOH was also associated with greater weekly alcohol consumption, (aOR 3.00; 95% CI 2.21 to 4.06 in those drinking 5–7 days/week compared with non-drinkers) and with increased alcohol consumption since the start of lockdown (aOR 1.34; 95% CI 1.14 to 1.69).

IPCOH was less likely among those reporting bad/very bad health (aOR 0.54; 95% CI 0.32 to 0.93). However, participants with COVID-19 experience (symptoms and/or diagnosis) were more likely to report IPCOH (aOR 1.34; 95% CI 1.10 to 1.65). Nearly one-third of participants scored highly on both the PHQ-2 and GAD-2 scores for depression and anxiety but after adjustment for age, gender, and relationship status, these mental health indicators were not associated with IPCOH (AOR 1.04; 95% CI 0.84 to 1.3 and AOR 1.08; 95% CI 0.86 to 1.34, respectively).

Figure 4 shows that in the 4 weeks prior to the interview, over a third of participants (38.4%) did not report any intimate contact, with this varying by age and relationship status. Most intimate contact occurred with someone living in the same household (51.8% overall). Of those reporting IPCOH (9.8%), two-thirds (64.4%, accounting for 6.3% of the whole sample) reported doing so with a person who was in their support bubble. The proportions reporting within-bubble rather than outside-bubble contact was similar across age groups. Over half of those in a steady relationship not living together reported IPCOH, with the majority (84.4%) reporting that this was with someone in their bubble.

Participants were asked to select up to three predefined reasons why they met partners outside their household. Nearly half (48.3%) gave 'I wanted to have sex' as a reason, with one-third selecting 'I missed them' (36.8%) and 'I was lonely and wanted some intimate physical contact' (34.1%) (table 2). 'I missed them' was the most common reason (reported by 75.9%) of those who were in a steady non-cohabiting relationship, with 'I wanted to have sex' the most common reason among the other relationship types.

Participants reporting IPCOH and included in our qualitative sample fell into two categories, those who were single (n=8) and those in steady non-cohabiting relationships (n=10). Distinct themes emerged within and between these two groups. Selected quotes, by theme are shown in table 3. The two groups differed in motivations and circumstances for IPCOH, but the accounts of both groups reflected complex and individualised decision-making that involved weighing up various different risks (such as SARS-CoV-2 transmission and judgement of peers) against benefits (such as feelings of security and improved mental health).

Participants in steady non-cohabiting relationships rarely discussed their motivations for IPCOH explicitly. Instead, they spoke about trading off social contact or applying tighter restrictions in other areas of their lives in order to maintain continuity in their relationship. Many rationalised seeing their partners as low risk or 'not doing any harm', positioning it in relation to other 'risks' (and associated behaviours) of becoming infected. Participants who were single described feelings of loneliness and boredom as key reasons for IPCOH, with contact providing security and human connection.

For both those who were single and those in steady relationships, there was evidence of deliberation in their decision making. Participants discussed thinking about potential consequences and often referred to government guidance, personal situations and COVID-19 risk. Fear of judgement or getting caught breaking the rules evoked feelings of guilt or embarrassment for some. Participants

**Table 1** Proportions, crude and age-adjusted ORs of reporting intimate physical contact in the past 4 weeks with a person who lives outside their household (IPCOH), in men and women aged 18–59 years in Britain (n=6654)

| Category | % of sample | % reporting IPCOH | 95% CI | Crude OR (95% CI) | aOR (95% CI)* | Unweighted, weighted |
|---|---|---|---|---|---|---|
| **All participants** | 100 | 9.9 | (9.1 to 10.6) | – | – | 6654, 6654 |
| **Demographic factors** | | | | | | |
| **Age group (years)** | | | | P<0.001 | P<0.001 | |
| 18–24 | 13.5 | 17.7 | (15.4 to 20.3) | 3.51 (2.74 to 4.50) | 3.51 (2.74 to 4.50) | 1046, 896 |
| 25–34 | 26.4 | 13.2 | (11.6 to 14.9) | 2.47 (1.96 to 3.11) | 2.47 (1.96 to 3.11) | 1911, 1753 |
| 35–44 | 24.0 | 8.0 | (6.7 to 9.5) | 1.41 (1.08 to 1.84) | 1.41 (1.08 to 1.84) | 1465, 1595 |
| 45–59 | 36.2 | 5.8 | (4.9 to 6.8) | 1.00 | 1.00 | 2232, 2410 |
| **Gender¶¶** | | | | P=0.024 | P=0.0088 | |
| Men | 49.8 | 10.9 | (9.8 to 12.1) | 1.00 | 1.00 | 3187, 3310 |
| Women | 49.9 | 8.8 | (7.9 to 9.8) | 0.80 (0.67 to 0.94) | 0.80 (0.67 to 0.94) | 3443, 3320 |
| **Ethnicity** | | | | P=0.7404 | P=0.0397 | |
| White† | 85.7 | 10.1 | (9.3 to 10.9) | 1.00 | 1.00 | 5837, 5593 |
| Asian/Asian British‡ | 8.1 | 8.6 | (6.2 to 11.8) | 0.83 (0.58 to 1.2) | 0.63 (0.44 to 0.91) | 395, 530 |
| Black/African/Caribbean/Black British§ | 3.4 | 8.8 | (4.8 to 15.6) | 0.86 (0.45 to 1.65) | 0.63 (0.33 to 1.23) | 127, 221 |
| Mixed/multiple ethnic groups/other¶ | 2.8 | 10.7 | (7.0 to 16.2) | 1.07 (0.66 to 1.74) | 0.76 (0.46 to 1.25) | 169, 185 |
| **Sexual identity** | | | | P<0.001 | P<0.001 | |
| Heterosexual | 96 | 9.6 | (8.9 to 10.4) | 1.00 | 1.00 | 5762, 6291 |
| Gay or Lesbian | 1.8 | 19.5 | (15.3 to 24.6) | 2.28 (1.68 to 3.11) | 2.50 (1.82 to 3.45) | 326118 |
| Bisexual | 1.4 | 16.9 | (13.3 to 21.1) | 1.91 (1.42 to 2.56) | 1.52 (1.12 to 2.05) | 393, 93 |
| Other | 0.8 | 14.9 | (8.1 to 25.9) | 1.65 (0.82 to 3.30) | 1.28 (0.58 to 2.81) | 74, 51 |
| **Region** | | | | P=0.28 | P=0.48 | |
| England | 86.7 | 9.6 | (8.9 to 10.4) | 1.00 | 1.00 | 5887, 5770 |
| Scotland | 8.6 | 11.7 | (9.1 to 14.9) | 1.24 (0.93 to 1.67) | 1.19 (0.89 to 1.60) | 509, 572 |
| Wales | 4.7 | 11 | (7.6 to 15.5) | 1.16 (0.77 to 1.75) | 1.09 (0.72 to 1.64) | 258, 312 |
| **Rurality** | | | | P=0.037 | P=0.19 | |
| Urban | 85.4 | 10.1 | (9.3 to 11) | 1.00 | 1.00 | 4895, 4896 |
| Rural | 14.6 | 7.8 | (6.2 to 9.8) | 0.75 (0.57 to 0.98) | 0.83 (0.64 to 1.10) | 846, 840 |
| **Education** | | | | P=0.68 | P=0.40 | |
| No qualification | 4.3 | 8.4 | (5.6 to 12.6) | 1.00 | 1.00 | 268, 283 |
| Below degree | 48.4 | 10.1 | (9.1 to 11.2) | 1.22 (0.77 to 1.92) | 1.22 (0.76 to 1.95) | 3195, 3221 |
| Degree or above | 47.3 | 9.8 | (8.7 to 10.9) | 1.17 (0.74 to 1.86) | 1.10 (0.68 to 1.75) | 3191, 3149 |
| **Social grade** | | | | P=0.083 | P=0.089 | |
| A upper middle class/B middle class | 22.6 | 9.7 | (8.3 to 11.3) | 1.00 | 1.00 | 1652, 1506 |
| C1 lower middle class/C2 skilled working class | 52.7 | 10.2 | (9.5 to 11.7) | 1.10 (0.90 to 1.35) | 1.17 (0.95 to 1.44) | 3442, 3508 |
| D working class/ E lower level of subsistence | 24.7 | 8.5 | (7.2 to 10.0) | 0.86 (0.67 to 1.10) | 0.94 (0.72 to 1.21) | 1560, 1640 |
| **Behavioural factors** | | | | | | |
| **Current relationship status** | | | | P<0.001 | P<0.001 | |
| Steady and living together** | 58.6 | 2.5 | (2.1 to 3.1) | 1.00 | 1.00 | 3827, 3889 |
| Steady and not living together** | 7.2 | 56.3 | (51.6 to 60.9) | 49.4 (37.2 to 65.5) | 43.9 (32.8 to 58.8) | 517, 475 |
| Casual/new†† | 4.9 | 36.5 | (31.2 to 42.2) | 22.0 (16.0 to 30.2) | 20.7 (15.0 to 28.4) | 341, 321 |
| Single | 29.4 | 8.9 | (7.6 to 10.3) | 3.72 (2.85 to 4.86) | 3.40 (2.58 to 4.46) | 1950, 1947 |
| **No of sexual partners in the past year‡‡** | | | | P<0.001 | P<0.001 | |
| 0 | 30.9 | 2.2 | (1.6 to 3) | 0.19 (0.14 to 0.28) | 0.18 (0.13 to 0.26) | 1663, 1721 |
| 1 | 59.0 | 10.3 | (9.3 to 11.4) | 1.00 | 1.00 | 3294, 3288 |

Continued

**Table 1** Continued

| Category | % of sample | % reporting IPCOH | 95% CI | Crude OR (95% CI) | aOR (95% CI)* | Unweighted, weighted |
|---|---|---|---|---|---|---|
| 2 | 5.5 | 34.4 | (29.1 to 40.2) | 4.57 (3.47 to 6.01) | 3.94 (2.95 to 5.27) | 336, 308 |
| 3+ | 4.7 | 43.2 | (37.2 to 49.3) | 6.61 (5.02 to 8.69) | 5.60 (4.19 to 7.46) | 345, 261 |
| **Condomless sex with a new partner in the past year‡‡** | | | | P<0.001 | P<0.001 | |
| No | 87.9 | 7.6 | (6.9 to 8.4) | 1.00 | 1.00 | 4863, 4861 |
| Yes | 12.1 | 33.3 | (29.7 to 37.1) | 6·06 (4.96 to 7.4) | 5.03 (4.07 to 6.21) | 724, 672 |
| **Days drinking alcohol in past week** | | | | P<0.001 | P<0.001 | |
| 0 | 37.2 | 5.8 | (4.9 to 6.8) | 1.00 | 1.00 | 2407, 2474 |
| 1–2 | 36.3 | 11.9 | (10.7 to 13.3) | 2.20 (1.78 to 2.73) | 2.16 (1.74 to 2.68) | 2466, 2417 |
| 3–4 | 16.6 | 12.4 | (10.5 to 14.6) | 2.31 (1.79 to 2.98) | 2.32 (1.79 to 3.01) | 1118, 1106 |
| 5–7 | 9.9 | 13.2 | (10.7 to 16.2) | 2.47 (1.84 to 3.32) | 3.00 (2.21 to 4.06) | 663, 657 |
| **Alcohol consumption since lockdown** | | | | P<0.001 | P=0.001 | |
| Decreased or remained the same | 79.2 | 9.1 | (8.4 to 10.0) | 1.00 | 1.00 | 5154, 5199 |
| Increased | 20.8 | 13.1 | (11.3 to 15.1) | 1.50 (1.24 to 1.81) | 1.34 (1.14 to 1.69) | 1417, 1364 |
| **Health-related factors** | | | | | | |
| **General health status** | | | | P=0.0028 | P=0.0284 | |
| Good—very good | 73.4 | 10.2 | (9.4 to 11.1) | 1.00 | 1.00 | 4846, 4870 |
| Fair | 21.1 | 10.1 | (8.6 to 11.9) | 0.99 (0.81 to 1.21) | 1.15 (0.93 to 1.41) | 1419, 1400 |
| Bad—very bad | 5.6 | 4.4 | (2.7 to 7.1) | 0.40 (0.24 to 0.68) | 0.54 (0.32 to 0.93) | 374, 370 |
| **COVID symptoms and/or diagnosis** | | | | P<0.001 | P=0.0046 | |
| No | 81.4 | 9.1 | (8.3 to 9.9) | 1.00 | 1.00 | 5358, 5413 |
| Yes | 18.6 | 13.3 | (11.4 to 15.4) | 1.53 (1.26 to 1.87) | 1.34 (1.10 to 1.65) | 1289, 1234 |
| **Symptoms of depression (PHQ-2)§§** | | | | P<0.001 | P=0.046 | |
| No | 70.9 | 9.0 | (8.2 to 9.9) | 1.00 | 1.00 | 4579, 4642 |
| Yes | 29.1 | 12.0 | (10.6 to 13.6) | 1.39 (1.16 to 1.66) | 1.20 (1.00 to 1.45) | 1964, 1902 |
| **Symptoms of anxiety (GAD-2)§§** | | | | P=0.0006 | P=0.0635 | |
| No | 71.2 | 9.1 | (8.2 to 10) | 1.00 | 1.00 | 4582, 4679 |
| Yes | 28.8 | 12.0 | (10.5 to 13.6) | 1.37 (1.14 to 1.63) | 1.19 (0.99 to 1.42) | 1988, 1889 |

All percentages are weighted.
Totals may not correspond to 100% due to rounding or participants not answering a specific question.
±38 participants did not answer the question on whether close contact was with someone in the same bubble, household or outside of both and 19 did not answer the question about current relationship status
*Age-adjusted ORs, adjusting for age as a continuous variable.
†White includes all those who identify as white English, Welsh, Scottish, Northern Irish, British, Irish, Gypsy or Irish Traveller, or from any other White background.
‡Asian includes those who identify as Indian, Pakistani, Bangladeshi, Chinese or from any other Asian background.
§Black includes those who identify as African, Caribbean or from any other black background.
¶Mixed ethnicity includes those who identify as white and black African, white and black Caribbean, white and Asian or any other mixed or multiple ethnic background.
**Refers to steady, married or civil partnership.
††Includes casual, new partner, end of relationship (eg, separating), 1 type of partner and 'other'.
‡‡Includes both opposite-sex and same-sex partners.
§§Participants were classified as having symptoms of depression or anxiety if they scored three or more on the patient health questionnaire two items (PHQ-2) or generalised anxiety disorder two items (GAD-2) scales.
¶¶Twenty-four participants who identified 'in another way' are included in data presented for all participants but excluded from 'men' and 'women'. Trans men and trans women are included in data for men and women, respectively.
aOR, age-adjusted OR; GAD-2, Generalised Anxiety Disorder (two items); PHQ-2, Patient Health Questionnaire (two items).

also referenced government guidance to justify behaviour or to note that they felt frustrated by them. Eight people mentioned bubbles specifically, although most of these used a definition of bubbles that was more flexible than official guidelines. Some described themselves as having numerous bubbles and others used bubbles to rationalise/justify IPCOH retrospectively, including for experiences before the policy was introduced.

## DISCUSSION
Physical distancing as an NPI to prevent SARS-CoV-2 transmission aims to reduce interactions between people

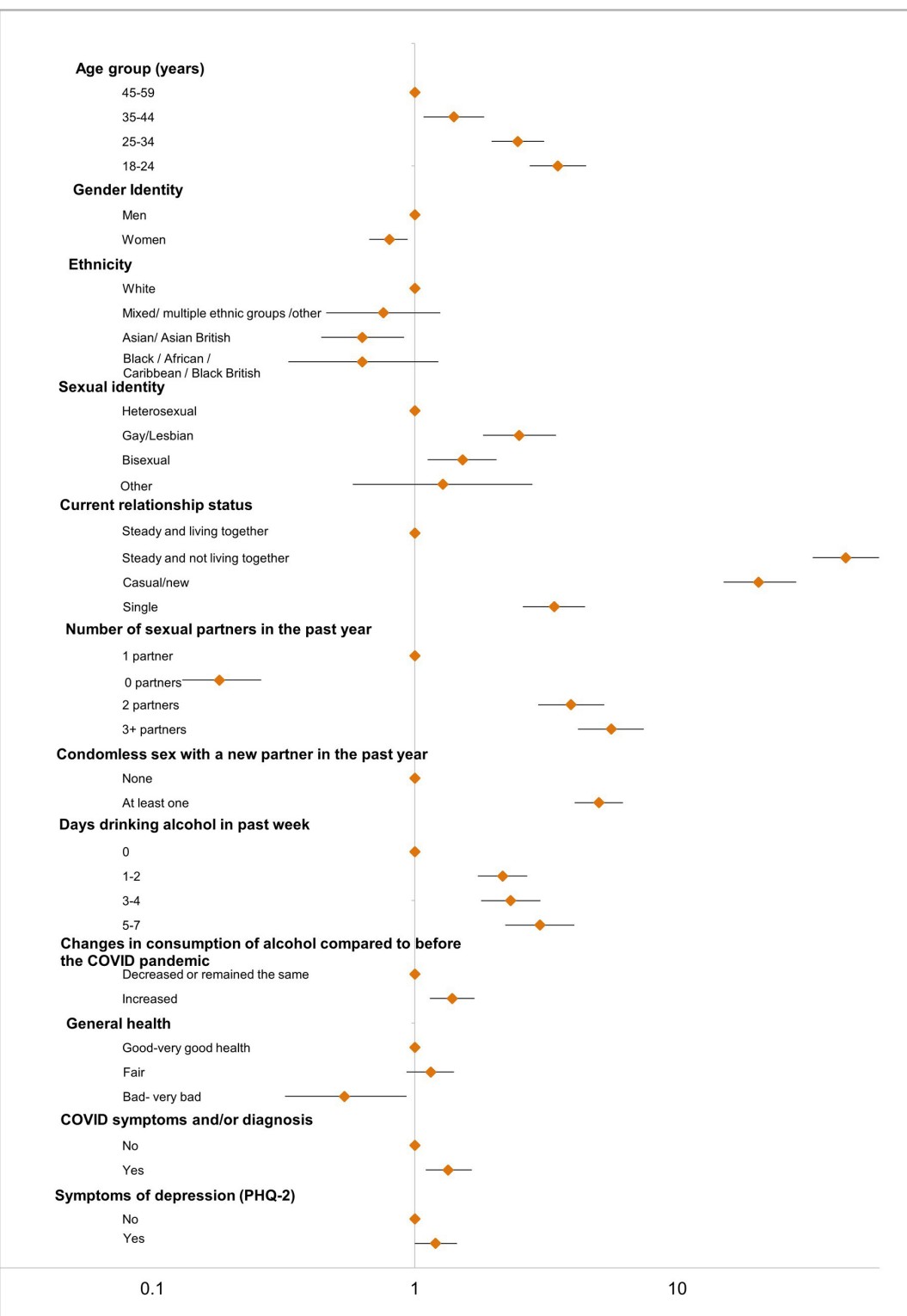

**Figure 2** Age-adjusted ORs for intimate physical contact in the past 4 weeks with a person who lives outside the household. PHQ-2, Patient Health Questionnaire (two items).

from different households. However, seeking sexual intimacy, among other factors, may affect adherence. Our research shows that nearly one in ten participants aged 18–59 reported IPCOH with a romantic/sexual partner in the 4 weeks prior to interview, (one in five in those aged 18–24), while the UK was still under some restrictions (figure 1). In June–July 2020, despite people in England being able to meet indoors or stay overnight if a one metre distance was kept, or people in Scotland and Wales being able to book a hotel or self-catering accommodation, sexual and other intimate contact with someone who lived outside the household was not permitted. Of

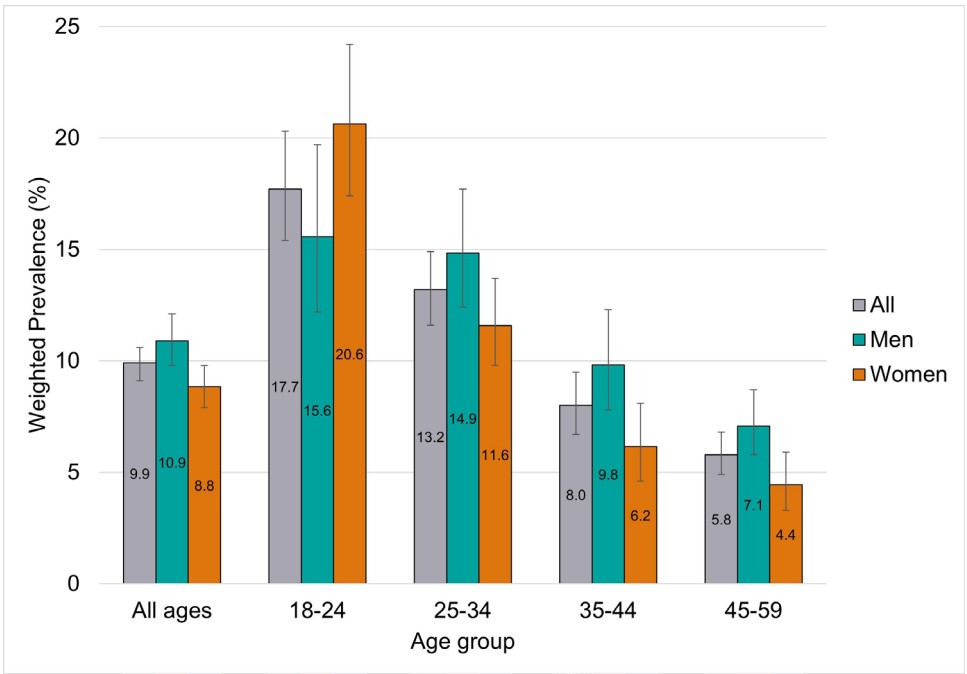

**Figure 3**  Intimate physical contact in the past 4 weeks with a person who lives outside their household, by age and gender (n=6654).

the individuals reporting IPCOH, most (86%) reported sex/genital contact, while a further 10% reported only kissing, which also carries a high SARS-CoV-2 transmission risk. We found that IPCOH was associated with other risk behaviours, such as higher numbers of sexual partners and condomless sex in the past year, and higher alcohol consumption, suggesting that some people who reported IPCOH were less risk averse more broadly.

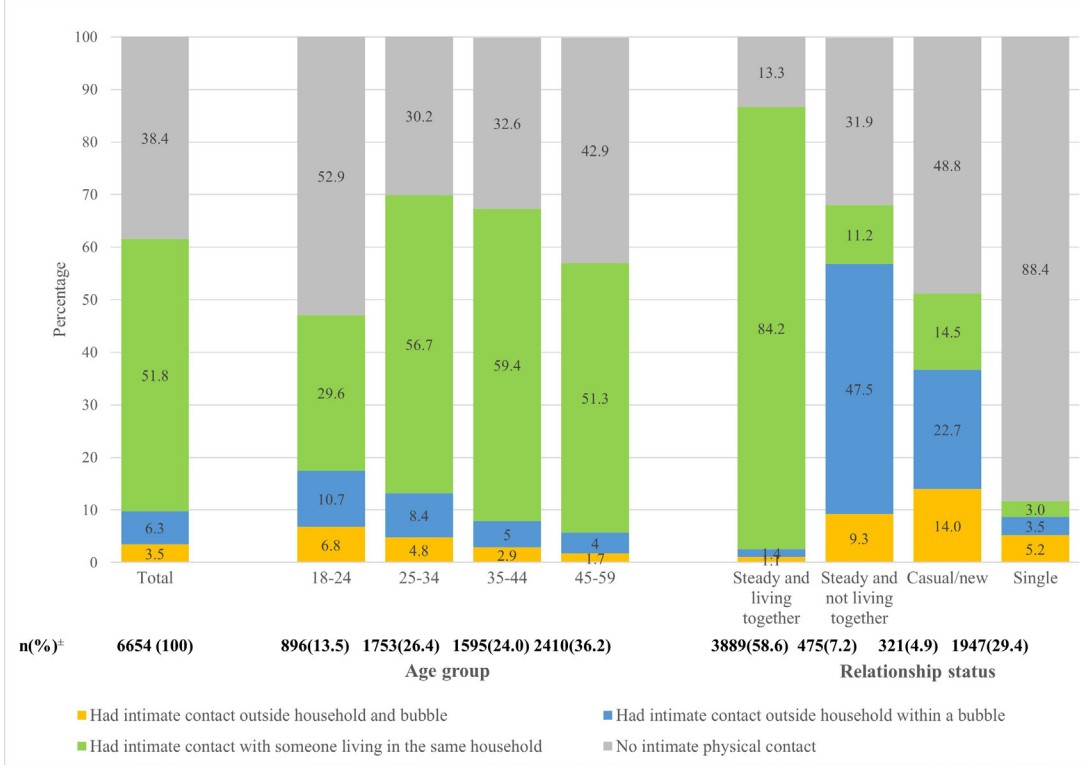

**Figure 4**  Intimate physical contact in the past 4 weeks with a person who lives outside their household and whether this was within a bubble or not, by age and relationship status. ±38 participants did not answer the question on whether close contact was with someone in the same bubble, household or outside of both and 19 did not answer the question about current relationship status. Therefore there are minor differences in the decimal places between this figure and Table 1

**Table 2** Reasons for intimate physical contact outside the household by relationship status*

| | Total | Steady and living together † | Steady and not living together † | Casual/new ‡ | Single |
|---|---|---|---|---|---|
| **Denominator (unweighted, weighted)** | **741, 656** | **116, 99** | **295, 267** | **140, 117** | **190, 172** |
| I missed them | 36.8 | 40.4 | 75.9 | 42.6 | 30.2 |
| I didn't want to lose contact | 14.9 | 19.2 | 17.3 | 15.9 | 18.4 |
| I was meeting up with them for another reason anyway | 11.1 | 5.8 | 16.9 | 13.1 | 12.1 |
| I was lonely and wanted some intimate physical contact | 34.1 | 33.6 | 16.3 | 46.4 | 34.0 |
| I wanted to have sex | 48.3 | 49.7 | 25.7 | 47.3 | 49.2 |
| I was bored and wanted some distraction | 15.3 | 24.4 | 2.5 | 11.7 | 17.5 |
| I needed to get away from my living place for a while | 20.3 | 25.0 | 21.5 | 14.3 | 14.3 |
| I was pressured into meeting them | 3.1 | 10.6 | 1.6 | 0.0 | 2.3 |
| Other | 7.3 | 3.9 | 14.5 | 7.5 | 6.7 |

**Colour legend for percentages**

| |
|---|
| 0 |
| 25 |
| 50 |
| 75 |
| 100 |

*Participants could tick up to three reasons.
†Refers to steady, married or civil partnership.
‡Includes casual, new partner, end of relationship (eg, separating), >1 type of partner and 'other'.

On the other hand, over 90% of participants did not report IPCOH and this includes nearly half of those in a steady non-cohabiting relationship. The qualitative interviews found that respondents did not embark on IPCOH lightly, rather they weighed up risks and benefits. Fear of judgement and risk of infecting both partners and others more generally were considered, suggesting that people took a societal view of infection that extended beyond the dyad. Respondents mitigated risks through reduction of social contacts, particularly with vulnerable people, in other areas of life. Respondents also deliberated who they told about IPCOH and rehearsed justifications in case they were questioned. Physical touch is a key aspect of intimacy and human experience, and being deprived of physical touch, which can take a heavy emotional toll, was not experienced equally across the population.[13] Thus, while some who reported IPCOH may be considered risk-takers, overall, they are risk-managers.

Qualitative data provide context and insight into motivations and considerations around IPCOH. However, there are limitations in the study design, timing and interpretation. Online data collection methods are often less representative, and we used quota sampling and weighting to address this. Comparison between Natsal, the decennial probability sample of sexual behaviour in Britain, with online panels found consistent differences in reporting

of sensitive behaviours.[24] Social desirability bias,[25 26] the pervasive national public health messaging, and social norms during the pandemic are likely to have resulted in underestimation of sensitive behaviours, including IPCOH, despite the measures in place to ensure confidentiality. On the other hand, as the survey took place once some restrictions had eased, this is likely to be a higher estimate compared with one measured early in lockdown.

There are challenges in reporting partnership type and cohabitation status, especially during a period when circumstances may change, as shown by people reporting IPCOH, with some defining themselves as single and others as in a new/casual relationship. Partnership typologies used in clinical practice or research[27] are less relevant when examining IPCOH as these are used to determine sexual risk, rather than COVID-19 risk behaviours, and do not emphasise cohabitation status.

Language, terminology and physical distancing restrictions shifted regularly since the initial lockdown, and there is a possibility that some participants may have misinterpreted some of the questions, despite definitions being provided to minimise this. This is exacerbated by differences in restrictions and public health messaging across the three nations and the timing of these interventions and the survey itself, as shown in figure 1. The

**Table 3**  Qualitative findings illustrating motivations for intimate physical contact outside the household

| Theme | Quote |
|---|---|
| **For those in steady non-cohabiting relationships** | |
| Continuity of contact | **…We just carried on as before(seeing partner, location and frequency)—We just carried on, sorry. (P1, F, 50–59)**<br>We've been going out for so long. We would see each other anyway and we were still talking all the time. So, it just made sense. (P17, F, 30–39) |
| About more than sexual needs | **To be honest we didn't really have sex that much. We're not highly sexed people in general, I think. So, it was more, we're more sort of being together with each other rather than shagging all of the time. (P5, F, 18–29)**<br>Where obviously I couldn't see her during lockdown she was getting extraordinarily depressed herself because she suffers with depression naturally anyway(…)So I'm—we waited for the first three weeks and when the government sort of came back with this, 'We'll announce a thing in three weeks' time,'(…)I think we waited that and when they come back with some other ridiculous decision we thought, 'Oh, we're not—I'm not standing for this anymore it's nonsense.' (P7, M, 50–59) |
| Weighing up risks and benefits | **So, we weighed up the risk of me catching it and passing it to him was very, very slim so we decided that was a risk we were OK with taking …I used to get parcels delivered to her(partner's grandmother's)house during the day because she would be in, but we had a discussion and we stopped getting parcels sent to her house. (P18, M, 18–29)**<br>To be honest when you say you're going to see somebody from a different household, I'm going to see somebody I love, I don't give a toss what you say. I'm not spreading anything. I don't give a toss. I'm washing my hands· Doing everything you've told me to do. It's no different from me going to bloody Morrison's(…)(Interviewer: Did you see any risks in meeting up with her?)Yeah, minimal. I'd gone through it in my head. One of my jobs is to go through risk. (P4, M, 30–39)<br>**Yeah. Well, you can get COVID from less than sex (P5, F, 18–29)**<br>Totally illegal I appreciate that but I think it's healthy for both of us to do so [to meet up] (P7, M, 50–59). |
| **For those who were single** | |
| Loneliness and boredom | **So like I said, I'm single, but during lockdown I thought it would be quite a good idea to start talking to people on dating apps. Because yes, I thought it would be quite a nice time to—well everyone I think was feeling a bit lonely. (P3, F, 30–39)**<br>**Yeah, I think it was another kind of coping mechanism that branched out I think, so I think that was like another, it got to me and I think I saw it as a way of like just maybe passing time. (P18, M, 18–29)**<br>Well, during lockdown I was bored and I just went on Tinder and got chatting to quite a few people then just to pass the time, that carried on for a couple of months…That's it really, yeah, just literally bored during lockdown and you run out of friends to talk to, run out of things to say to the kids. (P6, F, 40–49) |
| Unmet needs (emotional, sexual) | **There was this sense of no companionship, this sense of, you know, not feeling safe. A sense of physical intimacy not being there because physical intimacy is something that makes you feel safe and secure… [the sex] is something that you enjoyed, and that has a deep impact on you in terms of calming you down, I would say, and there was some reassurance in it. (P10, F, 18–29)**<br>But I will admit that a few times when I had time to leave the house, I did sort of break the rules and meet with a few guys because I sort of got a bit desperate. (P18, M, 18–29) |
| **Cross-cutting Themes** | |
| Deliberation in decision making | **I'm not seeing anybody else, so I'm not really putting anybody else at risk is what I thought· He can easily pass it on to me because he's going to work and on the buses and everything. But I wasn't in contact in any, you know, way, shape or form with my family, so I figured, we're not really doing anybody any harm (P1, F, 50–59, in a relationship)**<br>I lost my dad in April to COVID anyway, so I do have experience, but it's like, you know what, we're both being extra careful everywhere else. And we sort of, rightly or wrongly, feel we need to see each other, so we have been meeting.<br>(P9, F, 40–49, in a relationship)<br>**When I was living alone with her [housemate] there was this sense of, let's just take care of each other as long as we're stuck in the house together. So I wasn't sure…and it would still be okay if I was not living with another person(…)because putting another person at risk was a huge huge thing. (P10, F, 18–29, single)** |
| Fear of judgement | It wasn't as enjoyable as perhaps I felt it should have been mainly because I still had this guilt at the back of my head that we shouldn't be doing this· (P7, M, 50–59, in a relationship)<br>**I don't tend to talk about that sort of thing anyway, but I certainly didn't mention it to anyone because you don't know how seriously they were taking the restrictions, so you just don't want to mention that you're potentially breaking the rules by still meeting your partner before the bubbling was a thing, so it certainly wasn't anything I mentioned. (P16, M, 18–29, in a relationship)**<br>'Sod it, I'll come over to you,' so that's what we did. So I went over there and we did that. I was sort of—I would take her a bag of shopping with me for example pretending I was dropping her food off just in case I got stopped. (P7, M, 50–59, in a relationship) |
| Reference to government guidance | **Obviously, we were bubbling up before that was a thing, so we did still see each other. (P17, F, 30–39, in a relationship)**<br>And I think we could have justified it through the support bubble type thing but it's changed so many times actually I've lost track of where we are (P7, M, 50–59, in a relationship) |

questions on IPCOH were limited to the 4 weeks prior to interview in July/August 2020 and we did not capture the extent of IPCOH in the initial period of lockdown or whether this changed over time. Furthermore, the cross-sectional nature of the survey meant we were unable to determine temporality, for example, whether the association with COVID-19 symptoms/diagnosis was due to people being infected due to IPCOH or whether they decided to have outside household contact following a COVID-19 illness since they considered themselves at lower risk.

Guidelines also need to pre-empt unduly loose or critical interpretations. For example, messaging around social bubbles, intended to support isolated individuals, may have resulted in unintended consequences. Based on household composition, only a small proportion of adults live alone (or as a single adult with children)[28] who would be the group intended to benefit from the support bubble intervention. Across all ages, 6.3% of respondents reported IPCOH within a bubble (10.7% in those aged 18–24). It seems likely that many of these interactions were not compliant with government guidelines and that a liberal definition of 'bubbles' was used to rationalise and facilitate behaviour that is potentially a risk for SARS-CoV-2 transmission, although highly valued and desired by participants. A more upfront approach may balance sexual intimacy needs with measures to reduce transmission. For example, the Dutch National Institute for Public Health and Environment advised single people to identify a 'sex buddy', so that they could meet the same person for physical and sexual contact during lockdown and together plan how they would limit contact with others.[29] Government coronavirus guidance was referenced in differing ways in the qualitative interviews, with some indicating frustration that the published guidance did not adequately cover their relationship circumstances, while others used this as a reference point in their deliberations. There may also be misconceptions about the risk and routes of SARS-CoV-2 transmission, including related to sexual behaviour.[30]

The role that IPCOH plays in increasing the risk of SARS-CoV-2 transmission and expanding transmission networks is difficult to quantify, as romantic/sexual behaviours are only one part of broader social behaviours. A review of studies of contact patterns found that during initial lockdowns with stringent physical distancing measures there was a mean of 2–5 contacts per person per day, a 65%–87% reduction compared with pre-COVID rates, mainly due to reductions in outside household contact.[4] With low mean numbers of outside household contact, a single contact for sexual purposes may contribute a large proportion of overall contact numbers. However, much of IPCOH is within non-cohabiting steady relationships, where contact on repeated occasions is with the same person. It is worth noting that some participants described minimising other social contacts as a means to manage risk and enable or justify their intimate partner relationships.

Mathematical models that parameterise household structure and age-specific mixing patterns have estimated the contribution of bubbles on transmission.[31 32] However, these do not explicitly model the impact of bubbles in the context of IPCOH. While bubbles that comply with the definition (ie, single person households or those with one adult and children), are not predicted to have a large impact on transmission, this is not likely to be the case when a large proportion of the whole population are in bubbles, or when there is multiple household occupancy in both households in the bubble.[31] Furthermore, a lack of adherence to the exclusivity of bubbles could lead to rebuilding of contact networks that in turn lead to the epidemic threshold being crossed.[32] The impact on transmission is also dependent on level of transmission in the population at the time, and whether allowing bubbles affects wider risk perception in the population.

On a population level, IPCOH with a romantic/sexual partner may not be a large additional contributor to overall transmission risk. In our survey, we do not know the household size or whether those reporting IPCOH did so with the same person throughout lockdown, or more than one person (or household). There may be subgroups of the population where IPCOH plays a larger role in transmission, particularly where there is contact with multiple people from different households either at the same time or sequentially. While IPCOH was higher in some groups (eg, >50% in those in steady non-cohabiting relationships and 20% in people identifying as gay or lesbian), these groups constitute small proportions of the total population. Unlike sexually transmitted infections, where core groups and bridging populations contribute disproportionately to transmission dynamics,[33] this may not be the case for a highly transmissible respiratory pathogen that is distributed more homogeneously.

The COVID-19 pandemic has impacted sexual behaviour and relationships, with NPIs drastically reducing the opportunity to have sex for those not living with a partner.[12] This includes a number of domains of social relationships: social networks; social support; social interaction and intimacy.[13] People in non-cohabiting relationships, who are more likely to be younger,[26] bear much of the burden of restrictions on sexual/romantic interhousehold mixing. Young people are also less likely to develop severe COVID-19 disease, which means the risk–benefit of complying with physical distancing may be regarded as less favourable. They are also more likely to have suffered from depression and anxiety during the pandemic.[34] The study from Italy, that specifically focused on the impact of the pandemic on sexual and mental health, found that subjects who could maintain sexual activity during lockdown had lower psychological distress than those who had to give up on sexual activity due to lockdown policies.[14] IPCOH was also higher in sexual minorities. Interventions aiming to reduce SARS-CoV-2 transmission need to be cognisant of increasing stigma, potentially adding to inequalities and exacerbating existing poorer sexual and mental health outcomes.

Public health messages need to recognise sex and relationships, and the heterogeneity of circumstances. Slogans such as 'Hands, Face, Space' may convey a concise message, but do not reflect the complexity and nuance of people's households and relationships and how these affect decision making and adherence to control measures. NPIs and messaging also need to be updated as evidence accumulates on the relative contribution of different modes of SARS-CoV-2 transmission. As the pandemic progresses, factors such as the need for intimacy, connection, relationship maintenance and sexual fulfilment need to be weighed up against risk of SARS-CoV-2 transmission.

**Author affiliations**
[1]Institute for Global Health, University College London, London, UK
[2]Institute of Health Informatics, University College London, London, UK
[3]MRC/CSO Social and Public Health Sciences Unit, University of Glasgow, Glasgow, UK
[4]NatCen Social Research, London, UK
[5]Faculty of Public Health and Policy, London School of Hygiene and Tropical Medicine, London, UK
[6]Department of Social and Environmental Health Research, London School of Hygiene and Tropical Medicine, London, UK
[7]Department of Laboratory Medicine, Örebro University, Orebro, Sweden

**Acknowledgements** We want to thank the study participants and Margaret Blake and Reuben Balfour (Ipsos MORI). We thank Dr Caisey Pulford, Senior Surveillance and Prevention Scientist at Public Health England (now UK Health Security Agency), who designed a Figure that we used to create figure 1.

**Contributors** The paper was conceived by PS and NF, with further discussions with DM, LF, CHM, AMJ and KJM. PS wrote the first draft, with further contributions from DM, LF, KJM, DR, SC, CT, AC, JR, ED, RBP, JG, M-CR, WM, MU, CB, AMJ, CHM, KM and NF. Statistical analysis was done by DM, with support from PS, AC, ED and NF. Qualitative interviews were undertaken by KJM, DR and RBP. Coding and analysis of the qualitative data were undertaken by KJM and LF, with support from KJM. SC led on questionnaire design and data management was undertaken by JR and SC. PS and CHM are principal investigators (PIs) on Natsal and NF and KM are PIs on Natsal-COVID. All authors contributed to data interpretation, reviewed successive drafts and approved the final version of the manuscript. PS is the guarantor.

**Funding** Natsal is a collaboration between University College London (UCL), the London School of Hygiene and Tropical Medicine (LSHTM), the University of Glasgow, Örebro University Hospital, and NatCen Social Research. The Natsal Resource, which is supported by a grant from the Wellcome Trust (212931/Z/18/Z), with contributions from the Economic and Social Research Council (ESRC) and National Institute for Health Research (NIHR), supports the Natsal-COVID study in addition to funding from the UCL Coronavirus Rapid Response Fund and the MRC/CSO Social and Public Health Sciences Unit (Core funding).

**Disclaimer** The sponsors of the study had no role in study design, data collection, data analysis, data interpretation, or writing of the report.

**Competing interests** None declared.

**Patient consent for publication** Not applicable.

**Ethics approval** Ethical approval was obtained from the University of Glasgow MVLS College (reference 20019174) and LSHTM research ethics committees (reference 22565).

**Provenance and peer review** Not commissioned; externally peer reviewed.

**Data availability statement** Data are available in a public, open access repository. The Natsal-COVID study dataset has been deposited with the UK Data Archive with safeguarded access (SN 8865 - National Survey of Sexual Attitudes and Lifestyles COVID-19 Study, 2020). https://beta.ukdataservice.ac.uk/datacatalogue/studies/study?id=8865

**ORCID iDs**
Pam Sonnenberg http://orcid.org/0000-0002-1067-1583
Dee Menezes http://orcid.org/0000-0002-1628-1228
Karen J Maxwell http://orcid.org/0000-0002-2264-6510
David Reid http://orcid.org/0000-0001-6832-2418
Soazig Clifton http://orcid.org/0000-0002-4171-0805
Clare Tanton http://orcid.org/0000-0002-4612-1858
Andrew Copas http://orcid.org/0000-0001-8968-5963
Julie Riddell http://orcid.org/0000-0002-8084-4566
Emily Dema http://orcid.org/0000-0002-7254-2023
Raquel Bosó Pérez http://orcid.org/0000-0001-7342-4566
Jo Gibbs http://orcid.org/0000-0001-5696-0260
Mary-Clare Ridge http://orcid.org/0000-0001-9621-4529
Wendy Macdowall http://orcid.org/0000-0001-5868-8336
Magnus Unemo http://orcid.org/0000-0003-1710-2081
Chris Bonell http://orcid.org/0000-0002-6253-6498
Anne M Johnson http://orcid.org/0000-0003-1330-7100
Catherine H Mercer http://orcid.org/0000-0002-4220-5034
Kirstin Mitchell http://orcid.org/0000-0002-4409-6601
Nigel Field http://orcid.org/0000-0002-2825-6652

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
