## [Reviewer comments · BMJ Open]

ARTICLE DETAILS

TITLE (PROVISIONAL)	Intimate physical contact between people from different households during the COVID-19 pandemic: a mixed-methods study from a large, quasi-representative survey (Natsal-COVID)
AUTHORS	Sonnenberg, Pam; Menezes, Dee; Freeman, Lily; Maxwell, Karen J; Reid, David; Clifton, Soazig; Tanton, Clare; Copas, Andrew; Riddell, Julie; Dema, Emily; Bosó Pérez, Raquel; Gibbs, Jo; Ridge, Mary-Clare; Macdowall, Wendy; Unemo, Magnus; Bonell, Chris; Johnson, Anne; Mercer, Catherine; Mitchell, Kirstin; Field, Nigel

VERSION 1 – REVIEW

REVIEWER	Young, Honor Cardiff University, Centre for the Development and Evaluation of Complex Interventions
REVIEW RETURNED	25-Aug-2021

GENERAL COMMENTS	Thank you for providing such an interesting and relevant article to review. I thoroughly enjoyed reading it and fully support its publication. I have a few points only really to enhance the accessibility of the article. Introduction: Overall, the introduction provides some important information to set the scene for the study. However, I think it would benefit from a little more contextual information about the UK context; it would be useful to contextualise what issues are devolved across the UK for non-UK audience. Some more information more generally about the COVID 19 outbreak, timelines etc., might be of value to provide some additional information, that while it is current memory now, will (hopefully!) fade and we will need reminding. Some more specific points in relation to the text include the following: 1. “However, no study to date has measured the extent to which sexual or romantic relationships or encounters affect adherence to physical distancing measures” – given the temporal nature of the study is it possible to determine which is coming first? Perhaps this could be phrased as the relationship between, rather than affect.2. Is there a reference to support the transmission via connection of households: ‘people meeting for sexual or romantic reasons may expand transmission networks by connecting households.’3. The Figure 1 is lovely and provides a really clear visual description of how measures changed across Britain. I wonder if, again for context, little more information on what the ‘stringent measures’ were that were mandated when you say ‘Key dates include the announcement of the first national lockdown on 23 March 2020 when stringent measures were imposed.’ For example, what was involved with a national lockdown. I appreciate
---

	it's all fresh in our memories at the moment, but this will change over time. 4. What were the support bubbles referred to as in Wales? (referred to as "support bubbles" in England and "extended households" in Scotland. Could you also clarify that the 'guidance' mentioned in the next section is UK wide? Or rather came from devolved nations. 5. The wording 'IPCOH in the four weeks prior to interview in July-August 2020 and associated factors.' Makes it a little hard to determine the time frame of the interviews / when you are asking participants to reflect on. This could be clearer. 6. I appreciate the inclusion of the 'Average new diagnosed infections were ~4000 cases per day.¹⁵ But without additional information r.e. rates before / during / after lockdown this number is hard to interpret. 7. 'From early July, indoor mixing between households was allowed, albeit alongside physical distancing requirements.' Again, some more context here might be useful. Did this mean that everyone could meet inside? Were there restrictions in the numbers of people who were allowed to meet, and were they required to stay 2m apart (i.e. so really unless they were a bubble there really should have not been any IPCOH during the time that you are asking them to reflect on)? This could all be a little more explicit. Methods: 1. I appreciate the recruitment / sampling procedures for the quantitative part of the study are detailed elsewhere but it might be useful to have a sentence about recruitment using the online platform. For the qualitative work it would be useful to know what the inclusion criteria were for recruitment. (e.g. 'inclusion criteria, 18 respondents were included to fill age, gender, and ethnic background quotas, with the final sample characteristics as follows: male (n=9), female (n=9); 18-29 (n=5), 30-39 (n=5), 40-49 (n=4), 50-59 (n=4); white (n=14), from an ethnic minority group (n=4).) In addition, when it is said 'Respondents came from across Britain and regions of the UK under different levels of lockdown represented.' Would it be useful to have some information about the different levels of lockdown that were included at these points. 2. Could you define 'MVLS' when you discuss the ethical approval. 3. I'm not clear why the rationale for the analysis included IPCOH that was 'irrespective of whether the person considered this to be within a bubble or not.' Would bubbles not be considered extensions of households and therefore 'permitted/low risk of COVID transmission?') Results: 1. Would this be 'only kissing or holding hands' rather than 'kissing (10.3%) or only holding hands/hugging/cuddling (3.5%). 2. When you say 'Eight people mentioned bubbles specifically, although most of these used a definition of bubbles deviating from official guidelines' deviating from official guidelines, is this saying that they got the guidelines wrong/interpreted more flexible/strictly? It might be worth being more explicit if so. Discussion: 1. In the discussion, second sentence it says 'Seeking sexual intimacy' but there are other reasons relating to romantic relationships or encounters as outlined in the study. 2. You say 'while the UK was still under some restrictions. For example, at that time in England, people from two households could meet indoors and stay overnight, if a 1m distance was kept.' It's good to have an example of the restrictions but it might be
--	---

	worth explaining what the most strict / least strict restrictions were across the sampling frame just to give some context. 3. I'm not clear what is meant by 'timing' in the following sentence – is it referring to timing of lockdowns, easing of restrictions, or the survey? 'This is exacerbated by differences in restrictions, timing and public health messaging across the three nations (England, Scotland, and Wales) included in the survey, as shown in Figure 1' 4. I think I can understand what is being said here, but this could be a little more explicit (i.e. what is meant by those who are 'less able' to adhere, and which groups more likely to experience stigma). 'IPCOH was also higher in sexual minorities. Interventions aiming to reduce SARS-CoV-2 transmission need to recognise those who are less likely or able to adhere to lockdown measures and maintain physical distancing with those outside their household. However, these need to be cognisant of increasing stigma, potentially adding to inequalities and exacerbating existing poorer sexual and mental health outcomes.' Even more minor points: In the abstract Ipsos MORI should be capitalised as such, rather than IPSOS MORI. The text for the public involvement section seems to be a different size to the rest of the document. Thank you again for an interesting and pertinent article.
--	---

REVIEWER	Jannini , EA Universita degli Studi di Roma Tor Vergata, Department of Systems Medicine
REVIEW RETURNED	27-Sep-2021

GENERAL COMMENTS	The present manuscript focused on how restriction measures enacted during the COVID-19 pandemic in UK influenced the people's search for physical and sexual intimacy, giving the assumption that these behaviors could affect the adherence of prevention measures for SARS-COV-2 infection. The Authors aimed at exploring characteristics of people reported intimate physical contact outside the household (IPCOH) with the extent to find individuals' motivations and considerations around IPCOH. The very large sample and using the quota sampling as non-probability sampling method give a methodological and statistical strength to the research. Here are my comments: Introduction - You state that no research focused on IPCOH, and this is only partially right: One study (https://pubmed.ncbi.nlm.nih.gov/33234430/) stated that people separated from the partner during lockdown reported a worsen of psychological, relational and sexual health. Moreover, among separated partner, almost one quarter of them violated lockdown and social distancing measures in order to be sexually active nevertheless being not cohabiting with the partner. Please amend this section citing and discussing this article. Discussion - Even if you collect only qualitative data about motivation for IPCOH, you should discuss about your evidence, with the support of published data of several articles you didn't report in the manuscript. Here are a shortlist I suggest you: 1. doi: 10.1016/j.jsxm.2020.10.008.
--

	2. doi: 10.1136/jech-2021-216690. 3. doi: 10.1016/j.esxm.2020.100301. 4. doi: 10.1080/0167482X.2020.1807932. Finally, please check if percentages in total and IPCOH columns are correct in Table 1. In several rows the sum did not correspond to 100%.
--	--

VERSION 1 – AUTHOR RESPONSE

We would like to thank the reviewers for their positive response to the paper and for their helpful suggestions which we feel substantially improve the paper. We respond to each point and have edited the manuscript accordingly.

VERSION 2 – REVIEW

REVIEWER	Young, Honor Cardiff University, Centre for the Development and Evaluation of Complex Interventions
REVIEW RETURNED	17-Dec-2021
GENERAL COMMENTS	Thank you for addressing my comments so clearly and I look forward to seeing the paper published.